# Association between PDE4D rs966221 and the Risk of Ischemic Stroke in Regional Chinese Populations

**DOI:** 10.3390/brainsci13071038

**Published:** 2023-07-07

**Authors:** Chunyang Wang, Fenghe Zhang, Xiaojing Zhang, Chao Zhang, He Li

**Affiliations:** 1Department of Scientific Research, Tianjin Medical University General Hospital, Tianjin 300052, China; 2Department of Neurology and Institute of Neuroimmunology, Tianjin Medical University General Hospital, Tianjin 300052, China; 3Clinical College of Neurology, Neurosurgery and Neurorehabilitation, Tianjin Medical University, Tianjin 301700, China; 4Department of Neurology, Tianjin Neurological Institute, Tianjin Medical University General Hospital, Tianjin 300052, China; 5Tianjin Key Laboratory of Cerebrovascular and of Neurodegenerative Diseases, Department of Neurology, Tianjin Huanhu Hospital, Tianjin 300350, China

**Keywords:** ischemic stroke, genome-wide association study, pde4d, rs966221, Chinese population

## Abstract

In published research that includes genome-wide association studies and meta-analyses, the phosphodiesterase 4D (PDE4D) rs966221 variant has been identified as a risk factor in ischemic stroke (IS) in the Caucasian population. Several studies have investigated the relationship between rs966221 and IS susceptibility in Chinese populations over the years but have not provided consistently conclusive results. Therefore, our team performed a new meta-analysis of 5973 IS patients and 6204 controls from qualified studies. We observed no significant link between the PDE4D rs966221 variant and IS in any of the regional Chinese populations. Thus, we performed a subgroup analysis by the geographical distribution of China. Notably, significant associations were observed between rs96622 and the susceptibility of IS in the Northeast Chinese populations (*p* = 1.00 × 10^−4^, odds ratio = 1.28, and 95% confidence interval = 1.13–1.44, I^2^ = 0%). However, rs966221 was not found to be correlated with IS risk in the populations of North, Central, South, and East China. Our meta-analysis demonstrated that the PDE4D rs966221 variant is significantly associated with IS risk in some regional Chinese populations.

## 1. Introduction

Stroke is the second most important cause of death and the third most common cause of disability worldwide. According to the global statistics of stroke, ischemic stroke (IS) accounts for 62.4% of all types of stroke [1]. Because of the growing incidence of stroke, research on cerebrovascular diseases has turned its attention to the prevention and treatment of cerebrovascular diseases, particularly IS [2]. Ischemic stroke is a neurological condition with a complex etiology and a multitude of risk factors [3,4]. Genetic factors have been identified as major risk factors for IS [5]. 

Phosphodiesterase 4D (PDE4D) is one of the proteins of the phosphodiesterase family and participates in the hydrolyzation of cyclic adenosine monophosphate (cAMP) [6]. Research supports that PDE4 inhibitions have positive effects on neuroinflammation, neuroplasticity, and cognition [7,8,9]. In IS, cAMP plays an important role in many functions—predominantly neuron protection and post-ischemic inflammation regulation [10]. Researchers found that the ischemic brain has a lower cAMP level than the contralateral brain in the early three hours after the onset of IS [11]. It has been proved that increasing the signaling of cAMP in microglia could prompt a shift to an anti-inflammatory phenotype [12,13]. Xu et al. used the PDE4 inhibitor roflumilast to treat IS in middle cerebral artery occlusion (MCAO) rat models. The study found that roflumilast may lessen the neurological damage caused by cerebral ischemia–reperfusion injury by decreasing the infarct volume [14], and in the clinical trial, PDE4 inhibition has been reported to improve memory in healthy, old individuals [2]. 

The PDE4D rs966221 variant was first identified as a risk locus of IS in the Caucasian population in a genome-wide association study (GWAS) [15,16]. In the Pakistani population, Saleheen’s team found a significant link between rs966221 and IS [17]. Numerous studies in the Chinese population have consistently supported a substantial correlation between IS susceptibility and the PDE4D rs966221 variant [18,19,20]. However, conflicting findings suggest that rs966221 and IS risk are not significantly correlated in the Chinese population [21]. Besides the Chinese population, there is still no consensus on the question of whether rs966221 is related to susceptibility to IS [22]. Yoon et al. conducted a meta-analysis and did not find a link between rs966221 and the IS risk in the overall population [23], whereas the team of Yan obtained the opposite conclusion in the overall population [24]. Similarly, different studies about the association between rs966221 and IS have yielded inconsistent results in the Caucasian population [24,25,26].

Given these contradictory results, the relationship between rs966221 and IS in the Chinese population requires more study. To investigate the association between the frequency of the rs966221 genotype and the susceptibility of IS in the Chinese population, we performed a meta-analysis by combining data from previous case-control cohorts.

## 2. Materials and Methods

### 2.1. Systematic Literature Search

As a first step, we performed a systematic literature search using information from four databases: PubMed (http://www.ncbi.nlm.nih.gov/pubmed (accessed on 6 May 2023)), Google Scholar (https://scholar.google.com/pubmed (accessed on 6 May 2023)), the China National Knowledge Infrastructure (http://www.cnki.net/pubmed (accessed on 6 May 2023)), and the Wanfang Medicine database (http://www.wanfangdata.com.cn/pubmed (accessed on 6 May 2023)). We browsed all relevant studies using the following terms: “ischemic stroke”, “PDE4D”, and “Chinese or China”. Literature published before 15 October 2022 was selected for this study. 

### 2.2. Study Selection

In the second step, studies were required to meet the following criteria: (1) were human case-control studies, (2) included analyses of the association between the rs966221 variation and IS, and (3) provided sufficient information to allow calculation of an odds ratio (OR) and a 95% confidence interval (CI) for the rs966221 genotype. Studies not meeting these requirements were excluded.

### 2.3. Data Extraction 

Third, the following data were independently extracted from studies by two investigators: (1) the name of the first author; (2) the year of publication; (3) the study population; (4) the number of IS cases and controls; (5) the genotype distribution of rs966221 in cases and controls; and (6) the OR with the 95% CI or the data used to calculate the OR and the 95% CI.

### 2.4. Statistical Analysis

Finally, several statistical analyses were conducted. The chi-squared test was used to calculate the Hardy–Weinberg equilibrium of rs966221 in each study to prove that there is no difference in theoretical genotype frequencies and actual genotype frequencies. The significance level was established at *p* < 0.001. 

The relationship between PDE4D rs966221 and IS was estimated using the additive genetic model (the C allele versus the T allele). 

We used Cochran’s Q test and the I^2^ statistic to conduct a heterogeneity test [27]. The Q statistic has k-1 degrees of freedom and follows the *χ*2 distribution, with k signifying the number of studies selected in the calculation. A Cochran’s Q test *p*-value of less than 0.1 indicates significant heterogeneity between studies. The statistic I^2^ (I^2^ = (Q − (k − 1))/Q × 100%) indicates the percentage of variation across studies caused by heterogeneity. I^2^ ranges between 0 and 100% (I^2^ = 0–25%, 25–50%, 50–75%, and 75–100%), with a higher percentage indicating a higher level of heterogeneity [27,28]. 

A random effect model was employed for meta-analysis when heterogeneity was high (*p* < 0.1 of the Q statistic and I^2^ > 50%), and a fixed effect model was utilized in the remaining cases. The statistical significance of the OR was examined using the Z test, and a *p*-value < 0.05 was deemed significant.

Subgroup analyses were performed according to the seven geographical distributions of the Chinese population (South, Southwest, East, Central, Northeast, North, and Northwest Chinese populations). The genetic clustering in different populations has a strong correlation with their geographical distribution, ethnicity, and family of languages [29]. A sensitivity analysis was conducted by excluding each study sequentially from the included studies to calculate the impact of each study on the pooled OR and associated *p*-value [28]. A funnel plot was used to assess publication bias. Additionally, Egger’s regression approach was used to test publication bias in selected papers [30]. The significance level was set at 0.1. All statistical calculations were performed using R (http://www.r-project.org/pubmed (accessed on 6 May 2023)).

## 3. Results

### 3.1. Literature Search

Using a systematic literature search process, we selected 37 articles from four databases. The search flow is shown in Figure 1. Initially, five articles were removed because of duplication and review. Fifteen articles were excluded because they did not evaluate the relationship between the rs966221 variant and IS or contained insufficient data to allow a calculation of the OR. Three articles were subsequently omitted because the study control groups failed to meet the Hardy–Weinberg equilibrium. For the meta-analysis, we ultimately selected 14 articles that included a combined total of 5973 patients and 6204 controls. Among these fourteen studies, seven studies had the conclusion that rs966221 was correlated with IS risk [18,19,20,31,32,33,34], whereas seven articles stated that rs966221 had no association with the susceptibility of IS [21,35,36,37,38,39,40]. In most of the original case-control studies, the distribution level of hypertension, diabetes, and smoking risk factors in the IS group was significantly higher than in the control group. However, other IS risk factors, including alcohol consumption and BMI, did not have significant differences between the groups. Table 1 lists the specific details of the studies.

### 3.2. Meta-Analysis

Because of the high heterogeneity of the rs966221 polymorphism within the studies, we used a random effects model to compute pooled ORs and 95% CIs. The meta-analysis results showed no significant relationship between PDE4D rs966221 and IS (*p* = 0.115, OR = 1.15, 95% CI = 0.96–1.37).

Given that the 14 studies included in the meta-analysis were from different regions in China, we performed a subgroup analysis by region. In terms of the seven official geographic regions of China (Northeast, North China, East China, South China, Southwest, Northwest, and Central China), we excluded a subgroup with multiple populations from North, Southwest, and Northwest China. Subgroups were populations from North, Central, South, East, and Northeast China. Data on populations in North China were insufficient to calculate the OR and 95% CI. We utilized a random effects model for calculation because of the high heterogeneity in the Central Chinese (I^2^ = 98.6%) and East Chinese (I^2^ = 53.4%) populations. The remaining two regional populations did not show significant heterogeneity (I^2^ = 0%) and were, therefore, assessed using a fixed effect model. In Northeast Chinese populations, the results suggested significant associations between rs96622 and IS (North China: OR = 1.28, 95% CI = 1.13–1.44, *p* = 1.00 × ^−4^). However, no significant association was observed between rs966221 and IS in East Chinese (OR = 1.21, 95% CI = 0.99–1.46, *p* = 0.058), South Chinese (OR = 1.16, 95% CI = 0.96–1.39, *p* = 0.12), and Central Chinese (OR = 0.94, 95% CI = 0.34–2.60, *p* = 0.91) populations. Figure 2 and Figure 3 provide more detailed findings from the meta-analysis and subgroup analysis.

### 3.3. Sensitivity Analysis and Publication Bias Analysis

A sensitivity analysis was conducted by sequentially removing studies each time to assess the impact of each study. Removing any one of the South and Northeast China subgroups did not significantly influence the overall relationship between rs966221 and IS (Appendix A). 

A funnel plot was used to calculate the publication bias of the included studies. The funnel plot had an inverted, symmetrical shape, as shown in Figure 4. A regression test statistically demonstrated that none of the 14 studies in the meta-analysis had significant publication bias (*p* = 0.83).

## 4. Discussion

A member of the cAMP-specific PDE4 subfamily, PDE4D is reported to be distributed in a variety of cells, including vascular smooth muscle and immune cells [41]. In the pathogenesis of IS, PDE4D can inhibit the proliferation and migration of vascular smooth muscle cells to cause arteriosclerosis [41]. Xu’s team used magnetic resonance imaging of the vessel wall to confirm that the variants of PDE4D rs966221 were significantly associated with IS recurrence in patients with intracranial atherosclerosis [42]. In IS, the inhibition of PDE4 has further been proven to suppress the immune response in the brain [11].

For the global population, the association between rs966221 and IS has not yielded consistent results, and the differences in different research results may be related to experimental design, sample size, ethnic differences, and environmental factors. In the Chinese population, the relationship between rs966221 and IS susceptibility has been reported in many studies. A study by Zhang et al. (2019) of 1,773 participants (881 IS patients and 892 normal controls) from West China supported that PDE4D rs966221 was associated with a higher risk of IS [33]. In a study of a young cohort in the Northern Henan Province, rs966221 was identified as a risk variant of IS [43]. Wang et al.’s 2012 study of 340 participants in East China (235 IS patients and 105 healthy controls) identified an association between rs966221 and IS and suggested that PDE4D rs966221 significantly increased the risk of IS [31]. However, Shao and colleagues’ evaluation of 776 study participants from Zhejiang Province indicated no association between rs966221 and IS [40]. Finally, researchers reported no relationship between PDE4D rs966221 and susceptibility to IS in the Southern Chinese population [35]. Previous studies have concluded that gene–environment and gene–gene interactions may impact IS susceptibility. For Caucasian and Asian populations, the differences between ethnic genetic information and living environment, such as lifestyles and cultural perceptions, have an important influence on genetic research [44]. The differences in the research results may be related to experimental design, sample size, and ethnic differences. 

For further investigation, we screened previous articles and performed a new meta-analysis that included 5,973 IS patients and 6,204 controls from 14 studies in which seven studies had the conclusion that rs966221 was correlated with IS risk, whereas others reached opposite conclusions. Computed using a random effect model, the meta-analysis reported no significant association between PDE4D rs966221 and IS in the Chinese populations. Notably, we conducted a subgroup analysis by seven geographical regions of China and identified a significant association between rs96622 and IS observed in the Northeast Chinese population. In 2017, the team of Wang performed a population-based large-scale national survey of stroke, including 480 687 individuals in 31 provinces from seven official geographic regions in China, and researchers demonstrated that Northeast China had the highest incidence and mortality of stroke and ranked second in prevalence [45]. For the Northeast and other regions of the Chinese populations, the genetic variability of the population is limited, and environmental factors may play an important role in the results, such as dietary habits, health concepts, and work schedules. To some extent, our study reflects the regional genetic differences of IS in China, and it is worth exploring that after rational grouping and analysis, applying genetic results consistent with clinical studies may play an important role in disease risk assessment.
**Limitation**

Although we conducted meta-analyses that are effective studies of the association between rs96622 and IS in regional China, our study has several limitations. First, after the strict screening, the amount of original literature included in this meta-analysis is slightly insufficient. Furthermore, the cities involved in the original studies were limited to regional China. If possible, we will integrate more comprehensive and larger data based on the geographical distribution of China to further reveal the relation between rs96622 and IS susceptibility.

## 5. Conclusions

In sum, our meta-analysis offers reliable evidence of the association between the PDE4D rs966221 variant and vulnerability to IS in regional Chinese populations and provides a new thought for early disease prevention.

## Figures and Tables

**Figure 1 brainsci-13-01038-f001:**
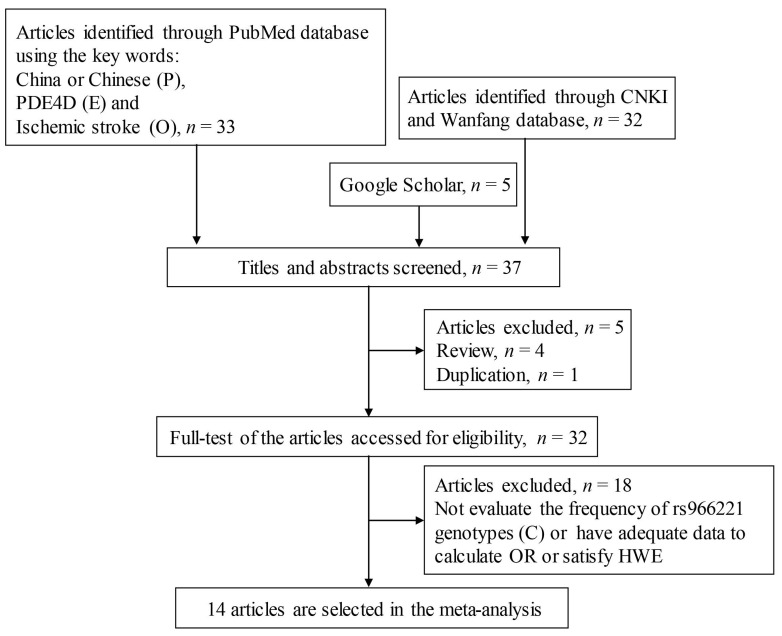
Flow of the literature search procedure. (P) Population, (E) Exposure, (C) Comparison, (O) Outcome.

**Figure 2 brainsci-13-01038-f002:**
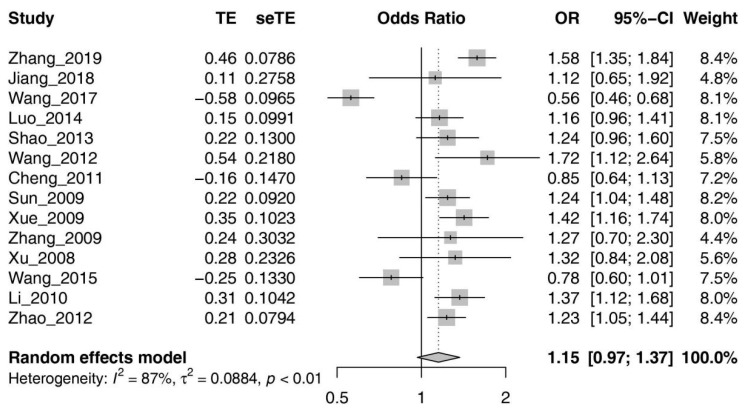
Forest plot for meta-analysis of the association between rs966221 and the susceptibility of IS. An OR equal to 1 indicates that the factor does not affect the occurrence of the disease; an OR greater than 1 (**right shift**) represents a risk factor; an OR less than 1 (**left shift**) indicates that the factor is a protective factor.

**Figure 3 brainsci-13-01038-f003:**
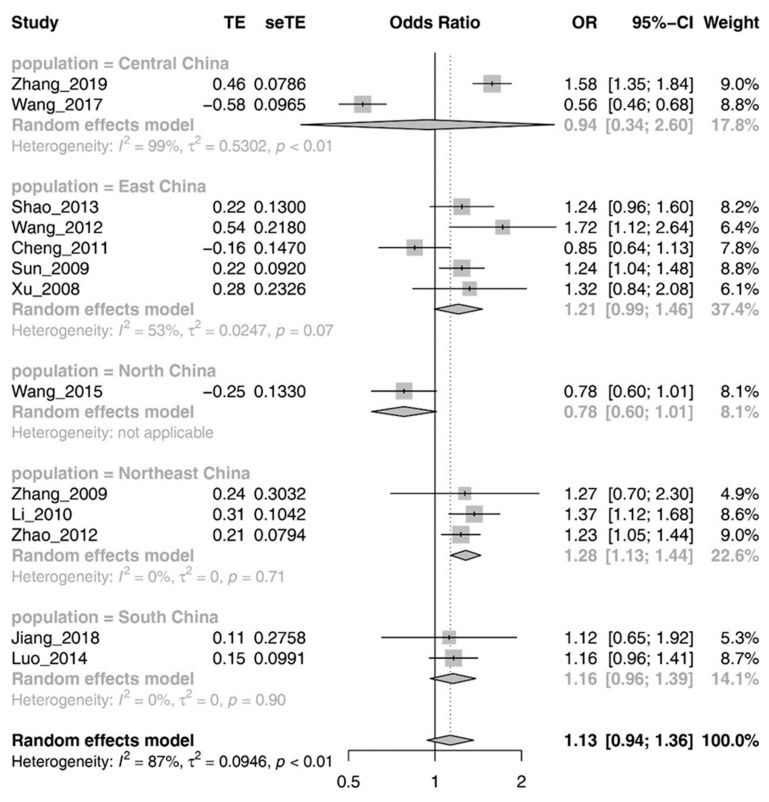
Forest plot of the result for subgroup analysis.

**Figure 4 brainsci-13-01038-f004:**
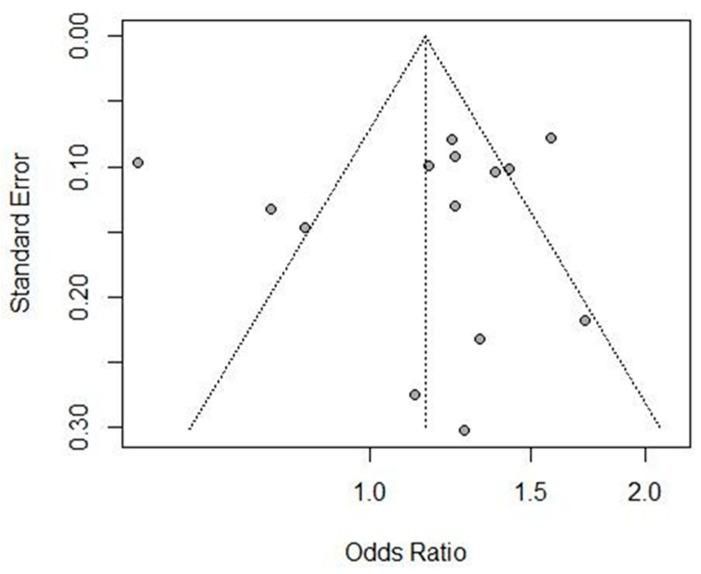
Funnel plot for publication bias analysis of the eligible studies evaluating the relationship between rs966221 polymorphism and the risk of IS. The *x*-axis and *y*-axis represent the ORs and standard errors for each eligible study, respectively.

**Table 1 brainsci-13-01038-t001:** Information of 14 eligible studies on the association between rs966221 and IS.

Study	Population/Province/City	Case	Control	HWE in Control	Case Genotypes			Control Genotypes		
					CC	CT	TT	CC	CT	TT
Zhang 2019 [33]	Central China/Hubei/Huangshi	881	892	0.308	86	341	454	43	282	567
Jiang 2018 [35]	South China/Guangxi/Chongzuo	101	104	0.228	1	30	70	4	22	78
Wang 2017 [32]	Central China/Anhui/ Hefei	610	618	0.501	312	240	58	410	182	24
Luo 2014 [36]	South China/Guangdong/Guangzhou	712	774	0.893	21	207	484	18	203	553
Shao 2013 [40]	East China/Zhejiang/Wenzhong	394	382	0.287	12	123	159	21	120	241
Wang 2012 [31]	East China/Zhejiang/Wenzhong	235	105	0.450	16	82	137	4	25	76
Cheng 2011 [21]	East China/Jiangsu/Nanjing	280	258	0.312	12	102	166	12	94	125
Sun 2009 [19]	East China/Shanghai	649	761	0.254	40	223	385	35	230	496
Xue 2009 [20]	Northwest, Southwest, and North China (Xi’an, Chongqing, Beijing, and Tianjin)	424	887	0.813	27	144	253	29	255	603
Zhang 2009 [39]	Northeast China/Heilongjiang/Harbin	122	44	0.067	7	46	69	4	10	30
Xu 2008 [38]	East China/Jiangsu/Nanjing	116	110	0.201	4	46	66	6	29	75
Wang 2015 [37]	North China/Beijing	396	300	0.324	13	123	260	19	100	181
Li 2010 [18]	Northeast China/Liaoning/Shenyang and Jinzhou	371	371	0.208	117	173	81	76	197	98
Zhao 2012 [34]	Northeast China/Liaoning/ Shenyang and Jinzhou	682	598	0.366	210	320	152	138	310	150

HWE: Hardy–Weinberg equilibrium; Genotypes of rs966221: CC, CT; TT.

## Data Availability

The original contributions presented in the study are included in the article/Appendix A, and further inquiries can be directed to the corresponding author.

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
