# Peer review of "Association between PDE4D rs966221 and the Risk of Ischemic Stroke in Regional Chinese Populations"

_brainsci, 2023, doi:10.3390/brainsci13071038_

Round 1
Reviewer 1 Report
This is a meta-analysis investigating the role of PDE4D rs966221 variant as a risk factor of ischemic stroke (IS) in all regional Chinese populations with a subgroup analysis by geographical distribution of China. Authors found that the PDE4D rs966221 variant is significantly associated with IS risk in the Northeast Chinese populations.
The methodology of this meta-analysis shoud be performed according to the Preferred Reporting Items for Systematic Reviews and Meta-Analyses (PRISMA) guidelines.
Author Response
Response to Reviewer 1
Thank you very much for your careful review of our manuscript and giving us these useful suggestions. We have carefully considered your comments. We accepted your suggestions and made several changes on the manuscript accordingly. All changes are marked in the revised version of the manuscript by using the “Track Changes”. We believe that the manuscript has been greatly strengthened according to your suggestions and hope that you will now find the paper suitable for publication.
Reviewer's Comment 1: The methodology of this meta-analysis shoud be performed according to the Preferred Reporting Items for Systematic Reviews and Meta-Analyses (PRISMA) guidelines.
Reply: Thank you for your suggestion. We have corrected and improved the content.
Reviewer 2 Report
This study performed a new meta-analysis of 5,973 IS patients and 6,204 controls from qualified studies. The authors observed no significant link between the PDE4D rs966221 variant and IS in all regional Chinese populations. However, significant associations were observed between rs96622 24 and the susceptibility of IS in the Northeast Chinese populations. Therefore, this study suggests that the PDE4D rs966221 variant is significantly associated with IS risk in some regional Chinese populations. This research paper studied an interesting topic with appropriate methodology. Here are my suggestions:
1. Meta-analysis should be based on a focused question ideally intended to solve a clinical problem. This question may be described in a PICO/PECO format (Patient/Problem/Population; Intervention/Exposure; Comparison and Outcomes). Please present this information.
2. Why is the Hardy–Weinberg equilibrium test performed? Please describe the purpose of this test in the method.
3. Please indicate the meaning of left and right shift of OR in figures.
4. 4. In the Caucasian population, ‘PDE4D rs966221’ was a risk factor for ischemic stroke, but this association was not seen in Asian Chinese. What are the mechanisms behind these conflicting results? Please add potential mechanisms to the discussion.
5. Only in the Northeast Chinese population was 'PDE4D rs966221' a risk factor for ischemic stroke, but not in other regions of China. What are the reasons for these differences?
6. Figure S1 seems to be one of the main findings, it seems preferable to switch to the main figure.
Author Response
Response to Reviewer 2
Thank you very much for your careful review of our manuscript and giving us these useful suggestions. We have carefully considered your comments. We accepted your suggestions and made several changes on the manuscript accordingly. All changes are marked in the revised version of the manuscript by using the “Track Changes”. We believe that the manuscript has been greatly strengthened according to your suggestions and hope that you will now find the paper suitable for publication.
Reviewer's Comment 1: Meta-analysis should be based on a focused question ideally intended to solve a clinical problem. This question may be described in a PICO/PECO format (Patient/Problem/Population; Intervention/Exposure; Comparison and Outcomes). Please present this information.
Reply: Thank you for your suggestion. We have corrected and improved the content. This question this study focused on was described in a PICO/PECO format: the association between the frequency of rs966221 genotype (Intervention) and the susceptibility of IS (Outcome) in the Chinese population (Population).
Reviewer's Comment 2: Why is the Hardy–Weinberg equilibrium test performed? Please describe the purpose of this test in the method.
Reply: Thank you very much for your suggestion. In the population, genotype frequencies consistent with the Hardy–Weinberg equilibrium proves that there is no difference in theoretical genotype frequencies and actual genotype frequencies. This test can be used to remove the SNP not conforming to this rule.
Reviewer's Comment 3: Please indicate the meaning of left and right shift of OR in figures.
Reply: Thank you very much for your suggestion. An OR equal to 1 indicates that the factor does not affect the occurrence of the disease; an OR greater than 1 (right shift) represents a risk factor; an OR less than 1 (left shift) indicates that the factor is a protective factor.
Reviewer's Comment 4: In the Caucasian population, ‘PDE4D rs966221’ was a risk factor for ischemic stroke, but this association was not seen in Asian Chinese. What are the mechanisms behind these conflicting results? Please add potential mechanisms to the discussion.
Reply: Thank you very much for your great opinion. The differences in research results may be related to experimental design, sample size, and ethnic differences. We add the content in the discussion.
Reviewer's Comment 5: Only in the Northeast Chinese population was 'PDE4D rs966221' a risk factor for ischemic stroke, but not in other regions of China. What are the reasons for these differences?
Reply: Thank you very much for your question. The genetic clustering of different subgroups of population has a strong correlation with their geographical distribution, ethnicity and family of languages. Usually, people go to the hospital within a relatively close range to their place of residence. Therefore, the results of different subgroups in this study to some extent reflect the genetic heterogeneity within the corresponding subgroup regions.
Reviewer's Comment 6: Figure S1 seems to be one of the main findings, it seems preferable to switch to the main figure.
Reply: Thank you very much for your suggestion. We have corrected the images in the reviewed manuscript.
Reviewer 3 Report
This study investigated the Association between PDE4D rs966221 and the Risk of Ischemic Stroke in Regional Chinese Populations, with a meta-analysis demonstrated that the PDE4D rs966221 variant is significantly associated with IS risk in some regional Chinese populations. The study is interesting. Due to conflicting results regarding the relationship between rs966221 and IS in the Chinese population, more studies were required.
Nevertheless, I have some comments and concerns, listed below:
Introduction
Lines 56-57: Concerning the study of Gretarsdottir et al. (2003), it would be interesting to give some details about this study and to add other research on the subject involving other countries, such as Pakistan, Italy or Brazil (references below), before developing those concerning China.
References for examples:
Saleheen, D., Bukhari, S., Haider, S. R., Nazir, A., Khanum, S., Shafqat, S., ... & Frossard, P. (2005). Association of phosphodiesterase 4D gene with ischemic stroke in a Pakistani population. Stroke, 36(10), 2275-2277.
Arnold, M. L., del Zotto, E., Lichy, C., Archetti, S., Werner, I., Padovani, A., ... & Grond-Ginsbach, C. (2006). PDE4D and stroke in the young. Genetic association of the SNP83T allele with ischaemic stroke in young stroke patients from Brescia (Italy) and Heidelberg (Germany). Aktuelle Neurologie, 33(S 1), P438.
da Silva, C. F., Schwartz, J., da Silva Belli, V., Ferreira, L. E., Cabral, N. L., & de França, P. H. C. (2020). Ischemic stroke and genetic variants: in search of association with severity and recurrence in a brazilian population. Journal of Stroke and Cerebrovascular Diseases, 29(2), 104487.
Material and Methods
· Statistical Analysis
- Line 95: It would be better to start the paragraph differently, such as "Finally, several statistical analyses were conducted. The chi-squared test was used to…”
Results
· Line 130-131-132: the authors selected seven studies got the conclusion that rs966221 was correlated with the IS risk (Sun et al., 2009; Xue et al., 2009; Li et al., 2010; Wang, 2012; Zhao et 131 al., 2012; Wang et al., 2017b; Zhang et al., 2019), but in Table 1 (Line 141). The reference Zhang et al. (2019) is missing.
Discussion
· The 14 studies included and cited in results should be discussed briefly, or grouped together to be cited in discussion.
· Line 194-195-196: The authors refer to “Wang et al.’s 2012 study of 340 participants in East China (235 IS patients and 105 healthy controls) identified an association between rs966221 and IS and suggested that PDE4D rs966221 significantly increased the risk of IS (Wang Hanmin 2012).”
ð (Wang Hanmin 2012) should be written (Wang et al., 2012).
· Line 196-197-198: The same remark can be made for the two following sentences: “However, 196 Shao and colleague’s evaluation of 776 study participants from Zhejiang Province indi- 197 cated no association between rs966221 and IS (Shao Minjie 2013).”
ð (Shao Minjie 2013) must be replaced by (Shao et al., 2013).
And Line 198-199-200: “Finally, researchers reported no relationship between PDE4D rs966221 and susceptibility to IS in the Southern Chinese population (Jiang Dongdong 2018).”
ð (Jiang Dongdong 2018) must be replaced by (Jiang Dongdong 2018).
References
· Line 329-330: The reference “Zhang, H.e.a. (2009). Study on association of the single nucleotide polymorphism of phosphodiesterase 4D with stroke. J 329 Chin Microcirc (06), 624-627.” must be adjusted:
ð Zhang, H. L., Wang, S. R., & Li, S. M. (2009). Study on association of the single nucleotide polymorphism of phosphodiesterase 4D with stroke. J Chin Microcirc, 13, 624-627.
· Line 291-292: The authors wrote “Wang, H.e.a. (2012). Association of Phosphodiesterase 4D Gene with Atherothrombosis Ischemic Strok. J Med Res 41(04), 291 134-137.”
This reference must be replaced by “Wang, H. M., Chen, X. L., Ye, H. H., Bi, Y., Pan, D. B., & Xu, L. Y. (2012). Association of phosphodiesterase 4D gene with atherothrombosis ischemic stroke. J Med Res, 41, 134-137.”
· Line 260-261: The authors wrote: “Jiang, D.e.a. (2018). Relationship between phosphodiesterase 4D gene rs966221 single nucleotide polymorphisms and is- 260 chemic stroke. Chinese journal of geriatric heart brain and vessel diseases 20(03), 271-274."
This reference must be replaced by “Jiang, D., LI, H., LI, J., Xiao, Y., Gan, L., & Luo, M. (2018). Relationship between phosphodiesterase 4D gene rs966221 single nucleotide polymorphisms and ischemic stroke. Chinese Journal of Geriatric Heart Brain and Vessel Diseases, 271-274.”
· Line 281-282: The authors wrote: “Shao, M.e.a. (2013). Correlation between ALOX5AP and PDE4D genes mutation and cerebral ischemic stroke in population 281 of Southern Zhejiang Province. (06), 592-595."
This reference must be corrected with the initials of the first name.
Funding:
· Line 227: The Funding section is not completed.
Comments on the whole manuscript
· Citations
- They should be listed in alphabetical order.
e.g.,: Lines 59-60: (Sun et al., 2009; Xue 59 et al., 2009; Li et al., 2010) => (Li et al., 2010 ; Sun et al., 2009; Xue 59 et al., 2009)
Line 131-132: (Sun et al., 2009; Xue et al., 2009; Li et al., 2010; Wang, 2012; Zhao et 131 al., 2012; Wang et al., 2017b; Zhang et al., 2019) => (Li et al., 2010; Sun et al., 2009; Wang, 2012; Wang et al., 2017b; Xue et al., 2009; Zhang et al., 2019; Zhao et 131 al., 2012)
Line 133-134: (Xu, 2008; Zhang, 2009; Cheng, 2011; Shao, 133 2013; Luo, 2014; Wang, 2015; Jiang, 2018)
- In a citation, it is appropriate to write “&” when the citation is in brackets. (Mika and Conti, 2016) => (Mika & Conti, 2016)
- Line 200: the citation (Jiang Dongdong 2018) is absent in References.
· The p-value should always be in italics, like all statistical values. Make the changes throughout the document.
e.g.,: Line 96: P < 0.001 => p < 0.001
Line 100: χ2 => χ2
· Table 1
- It is a good idea to consider displaying the table across the breadth of the page. But the table could be larger, so it may help with the line spacing.
- Spaces are missing very often before a bracket, especially when quoting authors.
e.g.,: Line 141, Table 1
Zhang 2019(Zhang et al., 2019) => Zhang 2019 (Zhang et al., 2019)
Line 192-193: In a study of a young 192 cohort in Northern Henan province, rs966221 was identified as a risk variant of IS(Yue et 193 al., 2019) => In a study of a young 192 cohort in Northern Henan province, rs966221 was identified as a risk variant of IS (Yue et 193 al., 2019).
The same remark is valid for all the citations in all this table.
- A note below the table is needed to explain the acronyms, not within the table.
Note. HWE: Hardy–Weinberg equilibrium; CC: = ? ; CT: = ?; TT: = ?
· Figures 2 and 3 are not of very good quality. The quality of pixels should be increased.
Author Response
Response to Reviewer 3
Thank you very much for your careful review of our manuscript and giving us these useful suggestions. We have carefully considered your comments. We accepted your suggestions and made several changes on the manuscript accordingly. All changes are marked in the revised version of the manuscript by using the “Track Changes”. We believe that the manuscript has been greatly strengthened according to your suggestions and hope that you will now find the paper suitable for publication.
Reviewer's Comment 1:
Introduction
Lines 56-57: Concerning the study of Gretarsdottir et al. (2003), it would be interesting to give some details about this study and to add other research on the subject involving other countries, such as Pakistan, Italy or Brazil (references below), before developing those concerning China.
References for examples:
Saleheen, D., Bukhari, S., Haider, S. R., Nazir, A., Khanum, S., Shafqat, S., ... & Frossard, P. (2005). Association of phosphodiesterase 4D gene with ischemic stroke in a Pakistani population. Stroke, 36(10), 2275-2277.
Arnold, M. L., del Zotto, E., Lichy, C., Archetti, S., Werner, I., Padovani, A., ... & Grond-Ginsbach, C. (2006). PDE4D and stroke in the young. Genetic association of the SNP83T allele with ischaemic stroke in young stroke patients from Brescia (Italy) and Heidelberg (Germany). Aktuelle Neurologie, 33(S 1), P438.
da Silva, C. F., Schwartz, J., da Silva Belli, V., Ferreira, L. E., Cabral, N. L., & de França, P. H. C. (2020). Ischemic stroke and genetic variants: in search of association with severity and recurrence in a brazilian population. Journal of Stroke and Cerebrovascular Diseases, 29(2), 104487.
Reply: Thank you very much for your suggestion. We have corrected the content in the reviewed manuscript.
Reviewer's Comment 2:
Material and Methods
- Statistical Analysis
- Line 95: It would be better to start the paragraph differently, such as "Finally, several statistical analyses were conducted. The chi-squared test was used to…”
Reply: Thank you very much for your suggestion. We have corrected the content in the reviewed manuscript.
Reviewer's Comment 3:
Results Line 130-131-132: the authors selected seven studies got the conclusion that rs966221 was correlated with the IS risk (Sun et al., 2009; Xue et al., 2009; Li et al., 2010; Wang, 2012; Zhao et 131 al., 2012; Wang et al., 2017b; Zhang et al., 2019), but in Table 1 (Line 141). The reference Zhang et al. (2019) is missing.
Reply: Thank you very much for your suggestion. We have corrected the content in the reviewed manuscript.
Reviewer's Comment 4:
Discussion
The 14 studies included and cited in results should be discussed briefly, or grouped together to be cited in discussion.
Line 194-195-196: The authors refer to “Wang et al.’s 2012 study of 340 participants in East China (235 IS patients and 105 healthy controls) identified an association between rs966221 and IS and suggested that PDE4D rs966221 significantly increased the risk of IS (Wang Hanmin 2012).”
ð (Wang Hanmin 2012) should be written (Wang et al., 2012).
Line 196-197-198: The same remark can be made for the two following sentences: “However, 196 Shao and colleague’s evaluation of 776 study participants from Zhejiang Province indi- 197 cated no association between rs966221 and IS (Shao Minjie 2013).”
ð (Shao Minjie 2013) must be replaced by (Shao et al., 2013).
And Line 198-199-200: “Finally, researchers reported no relationship between PDE4D rs966221 and susceptibility to IS in the Southern Chinese population (Jiang Dongdong 2018).”
ð (Jiang Dongdong 2018) must be replaced by (Jiang Dongdong 2018).
Reply: Thank you very much for your suggestion. We have improved the content in the reviewed manuscript.
Reviewer's Comment 5:
References
- Line 329-330: The reference “Zhang, H.e.a. (2009). Study on association of the single nucleotide polymorphism of phosphodiesterase 4D with stroke. J 329 Chin Microcirc (06), 624-627.” must be adjusted:
ð Zhang, H. L., Wang, S. R., & Li, S. M. (2009). Study on association of the single nucleotide polymorphism of phosphodiesterase 4D with stroke. J Chin Microcirc, 13, 624-627.
- Line 291-292: The authors wrote “Wang, H.e.a. (2012). Association of Phosphodiesterase 4D Gene with Atherothrombosis Ischemic Strok. J Med Res 41(04), 291 134-137.”
This reference must be replaced by “Wang, H. M., Chen, X. L., Ye, H. H., Bi, Y., Pan, D. B., & Xu, L. Y. (2012). Association of phosphodiesterase 4D gene with atherothrombosis ischemic stroke. J Med Res, 41, 134-137.”
- Line 260-261: The authors wrote: “Jiang, D.e.a. (2018). Relationship between phosphodiesterase 4D gene rs966221 single nucleotide polymorphisms and is- 260 chemic stroke. Chinese journal of geriatric heart brain and vessel diseases 20(03), 271-274."
This reference must be replaced by “Jiang, D., LI, H., LI, J., Xiao, Y., Gan, L., & Luo, M. (2018). Relationship between phosphodiesterase 4D gene rs966221 single nucleotide polymorphisms and ischemic stroke. Chinese Journal of Geriatric Heart Brain and Vessel Diseases, 271-274.”
- Line 281-282: The authors wrote: “Shao, M.e.a. (2013). Correlation between ALOX5AP and PDE4D genes mutation and cerebral ischemic stroke in population 281 of Southern Zhejiang Province. (06), 592-595."
This reference must be corrected with the initials of the first name.
Reply: Thank you very much for your suggestion. We have improved the content in the reviewed manuscript.
Reviewer's Comment 6:
Funding:
Line 227: The Funding section is not completed.
Comments on the whole manuscript
Citations
- They should be listed in alphabetical order.
e.g.,: Lines 59-60: (Sun et al., 2009; Xue 59 et al., 2009; Li et al., 2010) => (Li et al., 2010 ; Sun et al., 2009; Xue 59 et al., 2009)
Line 131-132: (Sun et al., 2009; Xue et al., 2009; Li et al., 2010; Wang, 2012; Zhao et 131 al., 2012; Wang et al., 2017b; Zhang et al., 2019) => (Li et al., 2010; Sun et al., 2009; Wang, 2012; Wang et al., 2017b; Xue et al., 2009; Zhang et al., 2019; Zhao et 131 al., 2012)
Line 133-134: (Xu, 2008; Zhang, 2009; Cheng, 2011; Shao, 133 2013; Luo, 2014; Wang, 2015; Jiang, 2018)
- In a citation, it is appropriate to write “&” when the citation is in brackets. (Mika and Conti, 2016) => (Mika & Conti, 2016)
- Line 200: the citation (Jiang Dongdong 2018) is absent in References.
- The p-value should always be in italics, like all statistical values. Make the changes throughout the document.
e.g.,: Line 96: P < 0.001 => p < 0.001
Line 100: χ2 => χ2
- Table 1
- It is a good idea to consider displaying the table across the breadth of the page. But the table could be larger, so it may help with the line spacing.
Spaces are missing very often before a bracket, especially when quoting authors.
e.g.,: Line 141, Table 1
Zhang 2019(Zhang et al., 2019) => Zhang 2019 (Zhang et al., 2019)
Line 192-193: In a study of a young 192 cohort in Northern Henan province, rs966221 was identified as a risk variant of IS(Yue et 193 al., 2019) => In a study of a young 192 cohort in Northern Henan province, rs966221 was identified as a risk variant of IS (Yue et 193 al., 2019).
The same remark is valid for all the citations in all this table.
- A note below the table is needed to explain the acronyms, not within the table.
Note. HWE: Hardy–Weinberg equilibrium; CC: = ? ; CT: = ?; TT: = ?
- Figures 2 and 3 are not of very good quality. The quality of pixels should be increased.
Reply: Reply: Thank you very much for your great opinion. We have improved the content in the reviewed manuscript.v
Round 2
Reviewer 2 Report
1. Adjust Figure 1 using the PICO/PECO format.
2. Add the meaning of the Hardy-Weinberg in the Methods.
3. Add the meaning of OR shift in Figure 3.
4. Add more rationale for the conflicting impact of rs966221 between Caucasian and Asian.
5. Add more rationale for the conflicting impact of rs966221 between Northeast and Other regions of China.
Author Response
Response to Reviewer 2
Thank you very much for your careful review of our manuscript and giving us these useful suggestions. We have carefully considered your comments. We accepted your suggestions and made several changes on the manuscript accordingly. All changes are marked in the revised version of the manuscript by using the “Track Changes” and in BLUE color. We believe that the manuscript has been greatly strengthened according to your suggestions and hope that you will now find the paper suitable for publication.
Reviewer's Comment 1: Adjust Figure 1 using the PICO/PECO format.
Reply: Thank you for your suggestion. We have corrected and improved the content. This question this study focused on was described in a PICO/PECO format: the association between the frequency of rs966221 genotypes (Comparison) of the PDE4D gene (Exposure) and the susceptibility of IS (Outcome) in the Chinese population (Population) and we have tried our best to adjust the Figure 1 following the PICO/PECO format based on the analysis process.
Reviewer's Comment 2: Add the meaning of the Hardy-Weinberg in the Methods.
Reply: Thank you very much for your great opinion. We have added related content in the reviewed manuscript.
Reviewer's Comment 3: Add the meaning of OR shift in Figure 3
Reply: Thank you very much for your great opinion. We have added related content in the reviewed manuscript.
Reviewer's Comment 4: Add more rationale for the conflicting impact of rs966221 between Caucasian and Asian.
Reply: Thank you very much for your great opinion. We have added related content in the reviewed manuscript.
Previous studies have concluded that gene-environment, gene-gene interactions may impact the IS susceptibility (Ferreira et al.). For Caucasian and Asian population, the differences between ethnicity genetic information and living environment like lifestyles, cultural perceptions have important influence on the genetic researches.
Reviewer's Comment 5:Add more rationale for the conflicting impact of rs966221 between Northeast and Other regions of China.
Reply: Thank you very much for your great opinion. We have added related content in the reviewed manuscript.
For the Northeast and other regions of Chinese, the genetic variability of population is limited, and environmental factors may play an important role in the results, such as dietary habits, health concepts, work schedule.
- Ferreira, L.E.; Secolin, R.; Lopes-Cendes, I.; Cabral, N.L. and França, P.H.C.Association and interaction of genetic variants with occurrence of ischemic stroke among Brazilian patients. Gene 2019, 695, 84-91, doi:10.1016/j.gene.2019.01.041